# O-GlcNAcylation in Gli1^+^ Mesenchymal Stem Cells Is Indispensable for Bone Formation and Fracture Healing

**DOI:** 10.3390/ijms26062712

**Published:** 2025-03-18

**Authors:** Moyu Liu, Yujie Hu, Chengjia You, Ding Xiong, Ling Ye, Yu Shi

**Affiliations:** 1State Key Laboratory of Oral Diseases and National Clinical Research Center for Oral Diseases, West China Hospital of Stomatology, Sichuan University, Chengdu 610041, China; liumoyu@stu.scu.edu.cn (M.L.); huyujie0222@foxmail.com (Y.H.); 2020224030001@stu.scu.edu.cn (C.Y.); xiongding@scu.edu.cn (D.X.); yeling@scu.edu.cn (L.Y.); 2Department of Endodontics, West China Hospital of Stomatology, Sichuan University, Chengdu 610041, China

**Keywords:** Gli1^+^ MSCs, hedgehog signaling, OGT, O-GlcNAcylation, bone formation, fracture repair

## Abstract

Adult mesenchymal stem cells (MSCs) play a crucial role in maintaining bone health and promoting regeneration. In our previous research, we identified Gli1^+^ MSCs as key contributors to the formation of most trabecular bone in adulthood and as essential for healing bicortical fractures. However, the mechanisms behind the maintenance and differentiation of Gli1^+^ MSCs are still not fully understood. O-linked N-acetylglucosamine modification (O-GlcNAcylation), mediated by O-GlcNAc glycosyltransferase (OGT), is involved in various biological processes and diseases. Our earlier work also demonstrated that O-GlcNAcylation is necessary for Wnt-stimulated bone formation. Nonetheless, the specific functions of O-GlcNAcylation in MSCs have not been completely elucidated. In this study, we found that the absence of OGT in Gli1^+^ MSCs led to a decrease in O-GlcNAcylation, which impaired both the bone formation and regeneration following fractures. Mechanistically, the Hedgehog signaling pathway induced O-GlcNAcylation through the insulin-like growth factor (Igf)-mTORC2 axis. This process stabilized the Gli2 protein at a specific site Ser355 and promoted osteogenesis in MSCs in vitro. Our findings reveal a significant mechanism by which O-GlcNAcylation regulates bone development and repair in mammals.

## 1. Introduction

The evolutionarily conserved Hedgehog (Hh) pathway is essential for normal embryonic development, as well as postnatal tissue growth, maintenance, renewal, and regeneration [1,2,3,4,5]. In mammals, there are three members of the Hh family: Sonic Hedgehog (Shh), Indian Hedgehog (Ihh), and Desert Hedgehog (Dhh). Hh proteins initiate signaling by binding to the receptor Patched (PTCH1), which mediates downstream signaling through Smoothened (SMO). This process leads to the transcriptional effector molecule Gli protein released from Kif7 and SUFU [6,7,8,9]. Additionally, small molecules, such as purmorphamine (PM), can artificially activate Hh signaling by targeting SMO [10]. Among the three Gli family members, the activation of Gli2, along with the repression of Gli3, are primarily responsible for Hh-induced gene activation. Gli1 plays a secondary role, serving to amplify the transcriptional response. Notably, we and others demonstrated that Gli1 is not only a downstream target of Hh signaling but also identified as an MSC marker that can label tissues, such as trabecular bone, articular cartilage, growth plates, teeth, periodontal tissues, and jaws [11,12,13,14,15]. Specifically, we have reported that in the metaphysis region, Hh-responsive cells, or Gli1^+^ MSCs, gave rise to most of the trabeculae [11]. Furthermore, among Gli1^+^ MSCs, chondrocyte-like osteoprogenitors were found to be sensitive to teriparatide during bone formation [16]. However, the detailed mechanism by which Gli1^+^ MSCs are involved in osteogenesis remains largely unknown.

O-GlcNAcylation is a unique post-translational modification of proteins [17,18] involving the synergistic action of OGT and O-GlcNAc hydrolase (OGA) [19]. This modification primarily targets serine or threonine residues, affecting the function, stability, and localization of proteins [20,21]. The existing literature demonstrated that a lack of OGT in osteoblasts, which resulted in reduced O-GlcNAcylation, led to a decrease in bone mass [22]. Our previous study also identified that O-GlcNAcylation was required for Wnt-stimulated bone formation through the modulation of glucose metabolism [23]. However, it remains uncertain whether O-GlcNAcylation plays a critical role in MSCs, particularly in Gli1^+^ MSCs during bone formation.

In this study, we demonstrated that the deletion of OGT adversely affects the functionality and differentiation of Gli1^+^ MSCs, ultimately leading to bone loss and repair defects in vivo. Mechanistically, the Hh pathway induces O-GlcNAcylation through the Igf-mTORC2 axis, which stabilizes the Gli2 protein at a specific O-GlcNAc site (Ser355) and facilitates the osteogenesis of MSCs in vitro. This work uncovered a hitherto unknown mechanism by which O-GlcNAcylation regulates mammalian bone development and repair.

## 2. Results

### 2.1. Lack of OGT in Gli1^+^ MSCs Decreased Trabecular Bone Formation in Postnatal Mice

To determine whether OGT is essential for bone formation in Gli1^+^ mesenchymal stem cells (MSCs), mice with the genotypes of *Gli1-CreERT2; OGT^+/Y^* (Ctrl) and *Gli1-CreERT2; OGT^c/Y^* (cKO) were treated with tamoxifen (TAM) via oral gavage at 4 weeks of age for five consecutive days and harvested 4 weeks later (Figure 1A). The micro-CT imaging and quantification of the cancellous bone in the distal femur revealed that the deletion of OGT in Gli1^+^ MSCs led to significant decreases in the bone mass (bone volume/total volume (BV/TV)), trabecular thickness (Tb.th), and trabecular number (Tb.N), along with decreased trabecular separation (Tb.Sp) (Figure 1B,C). However, measurements of the cortical bone thickness (Ct.Th) and cortical area (Ct.Ar) showed no significant differences (Figure 1B,C). This was consistent with previous findings where Gli1 primarily labeled MSCs in the chondro-osseous junction region [11]. The histological analysis using H&E and Masson staining confirmed a reduction in the trabecular structure in response to the OGT deletion (Figure 1D,E). To further investigate whether this trabecular defect was related to impaired bone formation, we conducted double-labeling experiments (Figure 1F). The quantification indicated that the deletion of OGT in the Gli1^+^ MSCs resulted in a decreased mineral apposition rate (MAR) in the trabecular bone, while there were no changes observed on the endosteal surface of the tibia (Figure 1G,H). Concurrently, the activity of the osteoclasts evidenced by TRAP staining and serum CTX-1 levels showed no differences between the Ctrl and cKO, indicating that the cancellous bone defects caused by the OGT deletion in the Gli1^+^ MSCs were due to the decrease in bone formation but not the alteration of bone resorption (Figure 1I–K).

### 2.2. OGT Depletion Led to the Depletion of the Gli1^+^ MSCs Pool

To test the underlying cause of the impaired bone formation, we traced the fate of the Gli1^+^ MSCs in vivo. The mice with the genotypes Gli1-CreERT2; OGT+/Y; Ai9 (Ctrl) and Gli1-CreERT2; OGTc/Y; Ai9 (cKO) were treated with TAM at 4 weeks of age for five consecutive days and harvested immediately (Figure 2A) or chased for one additional month (Figure 2H). Although the total number of Gli1^+^ MSCs did not change after this short period of OGT deletion (Figure 2D,G), the proliferation of these cells was dramatically decreased in the tibia of the cKO mice, as demonstrated by the EdU assay (Figure 2B,E). Furthermore, the osteogenic potential of Gli1^+^ MSCs was inhibited upon the OGT deletion, as indicated by the staining for osterix (OSX) (Figure 2C,F). In this context, the descendants of Gli1^+^ MSCs permanently expressed the red fluorescent protein (tdTomato), which we referred to as Tom+ cells [11,15]. Next, we examined the fate of Tom+ cells beneath the growth plate following OGT removal. The absence of OGT led to decreased proliferation (Figure 2I,L), which resulted in fewer Tom+ cells (Figure 2K,N). Importantly, the Tom+ cells that lacked OGT showed a lower tendency toward osteogenesis, as evidenced by decreased co-staining with the osteoblastic marker OSX (Figure 2J,M). This evidence supported the idea that the OGT deficiency decreased the maintenance and osteogenic differentiation of the Gli1^+^ MSCs, which was the primary cause of the impaired bone mass in the cancellous bone due to the OGT deletion.

### 2.3. OGT Deletion in Gli1^+^ MSCs Delays Fracture Repair

In our previous study, we found that postnatal mice recruited Gli1^+^ MSCs to form bone and cartilage callus during bone regeneration [11]. Unlike LepR^+^ cells, periosteal Gli1^+^ MSCs were demonstrated as the primary MSC source responsible for healing bicortical fractures [24]. To investigate whether the deletion of OGT in Gli1^+^ MSCs affects bone repair, we created a bicortical fracture in the femoral diaphysis of the mice. We first administered tamoxifen (TAM) to both control (Ctrl) and conditional knockout (cKO) mice at four weeks of age for five consecutive days. The fracture surgery was performed when the mice reached two months of age. We then harvested the mice on postoperative days (PODs) 14 (Figure 3A) and 21 (Figure 3H). Micro-computed tomography (micro-CT) analysis revealed that at POD 14, the mineralized calluses in the femurs of the cKO mice were smaller compared with their Ctrl littermates, as indicated by the lower bone volume/total volume (BV/TV) ratios and reduced bone mineral density (BMD) (Figure 3B–D). Histological analyses, namely, H&E staining, Masson staining, and Safranin O-Fast Green staining, showed an increase in the cartilage and a decrease in the bone percentage within the calluses of the cKO mice (Figure 3E–G). The bony calluses in the cKO mice remained smaller compared with the Ctrl mice, which was again reflected in the lower BV/TV and BMD measurements (Figure 3I–K). The histological results consistently demonstrated that the cartilage callus was still present in the cKO mice but was absent in the Ctrl mice, indicating a delay in the bony callus formation (Figure 3L–N). In summary, we concluded that the OGT was essential for the involvement of the Gli1^+^ MSCs in bone regeneration. A deficiency in O-GlcNAcylation hindered the repair process following the bone fractures.

### 2.4. Hh Modulated O-GlcNAcylation Through the Igf-mTORC2 Signaling Cascade

The data collected thus far indicate that the OGT in the Gli1^+^ MSCs was essential for normal bone formation and fracture healing in vivo. As the Hh-responsive cells, the Gli1^+^ MSCs were tightly regulated by the Hh pathway. Our previous work demonstrated a lack of Smoothened (SMO), which functioned as a key mediator to transmit Hh signals, caused the defects in the proliferation and differentiation of Gli1^+^ MSCs [11]. Additionally, Hh signals upregulated the expressions of Igf-1 and Igf-2 through SMO, which then activated the mTORC2 complex, subsequently protecting Gli2 from degradation [25]. Given these findings, we investigated whether the Hh pathway increased the O-GlcNAcylation through the Igf-mTORC2 axis. To achieve this goal, we treated the MSC line M2-10B4 with purmorphamine (PM), an agonist of Hh, to induce osteogenic differentiation for 7 days. We observed a significant increase in O-GlcNAcylation in response to PM. Furthermore, PM also upregulated protein levels of Gfat1, OGT, and OGA, indicating that the changes in O-GlcNAcylation were due to the modulation of regulatory enzymes in response to PM (Figure 4A). Meanwhile, the inhibition of SMO by Cyclophamine, a selective inhibitor of SMO, blocked PM-induced O-GlcNAcylation, together with the regulatory enzymes (Figure 4B). Furthermore, the mTOR pathway inhibitor Torin 1 dramatically reduced the OGT expression, which led to a blockage of O-GlcNAcylation, even in the presence of PM (Figure 4C). Consistently, the knockdown of Rictor, a unique component of mTORC2, significantly decreased the PM-induced OGT expression and O-GlcNAcylation (Figure 4D). Next, we observed the knockdown of Igf1r, which transmitted both Igf1 and Igf2 signals, also significantly suppressed the OGT protein level, which inhibited O-GlcNAcylation (Figure 4E). Collectively, these data suggest that the Hh pathway promoted O-GlcNAcylation via the mTORC2-Igf signaling axis.

### 2.5. O-GlcNAcylation of Gli2 Was Required for Hh-Induced Osteogenesis

We then asked whether O-GlcNAcylation was necessary in the Hh-induced osteogenesis in vitro. To this end, we treated the M2 cells with PM and introduced OSMI, a specific inhibitor of OGT. The suppression of O-GlcNAcylation using OSMI resulted in a downregulation of osteogenic differentiation, as evidenced by the decreased expression of the *Sp7* and *Alp* genes (Figure 5A). Additionally, there was a reduction in the osteogenic-specific staining, including ALP staining and Alizarin Red staining (Figure 5B). Next, we explored the mechanism by which O-GlcNAcylation mediates Hh-induced osteogenesis by examining its potential effects on Gli2, the primary regulator that activates the Hh pathway and increases Gli1 expression. To this end, we infected the M2 cells with lentiviruses expressing Flag-tagged full-length Gli2 (Flag-Gli2) in response to doxycycline (Dox). We then blocked protein de novo synthesis with cycloheximide (CHX) after inducing the Flag-Gli2 expression with Dox to determine the impacts of PM and OSMI on the half-life of Gli2. We monitored the abundance of Flag-Gli2 over time with treatments of PM and OSMI. Our data reveal that PM extended the half-life of Flag-Gli2 from approximately 6 to 24 h, while the OSMI eliminated the stabilizing effect of PM (Figure 5C,D), suggesting that there may be O-GlcNAcylation sites on the Gli2 protein. Analysis of the mouse Gli2 protein sequence using the YinOYang website revealed a potential O-GlcNAc site at serine 355 (S355). To determine whether S355 is a functional O-GlcNAc site on Gli2, we created a point mutation to change serine to alanine and generated a mutation construct S355A. We used the Flag fusion protein to analyze both the wild-type Gli2 (Gli2-WT) and the Gli2 mutant S355A (Gli2-S355A). The results indicate that the S355A mutation led to a significant decrease in the level of full-length Gli2 protein compared with the wild type (Figure 5E). Moreover, Gli2 with the S355A mutation exhibited a reduced half-life (Figure 5F,G), suggesting that S355 functioned as a critical site for Gli2 stability. Furthermore, our data show Gli2-S355A also impeded Hh-induced osteogenesis, as evidenced by the lower expression levels of *Sp7*, *Ibsp*, and *Alp* genes compared with Gli2-WT (Figure 5H). Thus, our data clearly demonstrate that S355 served as an O-GlcNAc site that stabilized the Gli2 protein, which, in turn, facilitated Hh-induced osteoblastogenesis.

## 3. Discussion

Hedgehog (Hh) signaling plays a crucial role in mammalian skeletal development. The Ptch receptor binds to the Hh ligand, leading to the deregulation of Smoothened (SMO) [26], which ultimately releases activated Gli2 as an effector that drives the transcription of target genes, such as Gli1 [27]. The Hh-Ptch-Smo-Gli axis is intimately involved in the regulation of osteogenic differentiation. Blocking the Hh pathway through the specific deletion of Ptch1 in HOC+ mature osteoblasts leads to severe osteoporosis [28], and the deletion of SMO in Osx+ or Gli1^+^ osteogenic progenitors also results in decreased bone formation [11,29]. Furthermore, the removal of the repression of Gli3 represents the principal means by which Hh governs chondrocyte proliferation and maturation. However, osteogenic differentiation and cartilage vascularization require the additional involvement of Gli2 activation for endochondral ossification, indicating that Gli2 is a key effector in the regulation of Hh-mediated osteogenic differentiation [30,31,32]. Additionally, Gli1 expression is not only the consequence of Hh signal activation but also reflects the Hh-responsive cells. Our group and others demonstrated that Gli1^+^ cells function as progenitors or MSCs at various skeletal sites, including postnatal Gli1^+^ cells in limb bones that contribute to both chondrocytes and osteoblasts during bone development and fracture healing [11,15]. Moreover, Gli1^+^ cells in periodontal ligaments support cementum growth, periodontal tissue metabolism, and the repair of damage [12,13]. However, it is unclear how Gli1^+^ MSCs are regulated in vivo.

O-GlcNAcylation, a post-translational modification of proteins, primarily involves three enzymes. Gfat1 is the first and rate-limiting enzyme of the hexosamine pathway, the catalytic product of which can serve as a substrate for uridine diphosphate N-acetylglucosamine (UDP-GlcNAc), which is the direct donor of O-GlcNAcylation. OGT catalyzes the addition of UDP-GlcNAc to serine or threonine residues, while OGA is responsible for removing UDP-GlcNAc. Together, both enzymes directly maintain the dynamic balance of intracellular O-GlcNAcylation levels [33]. In our previous work, we demonstrated that a decrease in O-GlcNAcylation in osteoblasts due to knocking down OGT impairs Wnt-induced bone formation via glycolysis rewiring [23]. Although we established a correlation between Hh-induced bone formation and O-GlcNAcylation here, whether O-GlcNAcylation is regulated by other signaling pathways to induce bone formation remains further studied.

Mechanistically, we propose that the Hh pathway induces O-GlcNAcylation through the IGF-mTORC2 axis, which stabilizes the Gli2 protein at a specific site and facilitates osteogenesis in MSCs in vitro (Figure 6). Previous studies support the notion that Gli2 serves as a central hub in the Hh signaling pathway [25]. Importantly, a mutation in Gli2 at a specific site resulted in decreased, but not completely blocked, O-GlcNAcylation, suggesting that other potential O-GlcNAc sites have yet to be identified. Additionally, since phosphorylation and O-GlcNAcylation compete for the same serine or threonine residues, it is unclear how these two processes are balanced [34]. Notably, mTOR has been reported to activate bone mass via the regulation of Igf1 [35]. And Igf2-mTORC2-Akt signaling is required for Hh-induced osteoblastogenesis [25]. Given the importance of mTOR in Hh-induced bone formation and the widespread occurrence of intracellular O-GlcNAcylation [36], other molecules in the mTOR and Hh signaling pathways, such as SMO, may also undergo O-GlcNAcylation. A deeper understanding of the regulatory mechanisms governing the O-GlcNAcylation of signaling pathway molecules in the skeletal system could help identify the universal regulators of O-GlcNAcylation, thereby enabling the precise regulation of osteogenic differentiation.

## 4. Materials and Methods

### 4.1. Mice

The mouse strain *Gli1-CreERT2*; *OGT^+/Y^*; *Ai9* was as described in [11,23]. All the mice were raised in a special pathogen-free animal facility under 12 h of light daily and provided with standard breeding feed and sterilized drinking water. The Animal Care and Use Committees of the West China Hospital of Stomatology (No. WCHSIRB-D-2022-548). approved all the mice.

### 4.2. TAM Administration

To induce the tdTomato expression and OGT knockout in the Gli1^+^ cells, the Gli1-CreERT2; OGT+/Y; Ai9 mice were administrated TAM (APE×BIO (Houston, TX, USA), C4624) (4 mg/30 g body weight, dissolved in corn oil) at 4 weeks of age for 5 consecutive days. The experiments involved subjects of both sexes, and no sex-dependent disparities were detected. Puppies of the same sex from the same litter were used in comparison experiments.

### 4.3. Immunofluorescence Staining

The immunofluorescence staining protocol was conducted as follows. Briefly, isolated mouse tibiae were soaked in 4% paraformaldehyde (PFA) on a shaker for 16 h, then turned to 14% EDTA decalcification buffer (pH = 7.2) for three consecutive days. Afterward, the tibiae were soaked in 30% sucrose overnight to dehydrate, and then they were embedded in an optimal cutting temperature compound (Epredia (Shanghai, China)). Then, 10 µm frozen sections were prepared using a Leica cryostat furnished with Cryojane (Leica (Wetzlar, Germany), CM1950). All steps were performed at 4 °C. The solutions used were more than 10 times the volume of the tissue. IF primary antibodies employed in this research included Osx/Sp7 (Abcam (Cambridge, UK), ab22552, 1:100). The secondary antibody was Alexa Fluor 488 (Invitrogen (Carlsbad, CA, USA), A11008, 1:200). Immunofluorescence-stained frozen tissue sections were imaged using a laser scanning confocal microscope (Olympus (Tokyo, Japan), FV3000, 10× objective lens).

### 4.4. Bone Fracture

Two-month-old male mice were used for fracture injury modeling, as previously described [37]. Briefly, mice were anesthetized with avertin and then incised along the skin and muscles. The femur was exposed through blunt dissection and fixed by an intramedullary pin. The cortical bone was then cut down in the midpoint, and sterilization and analgesia were performed after suturing the incision.

### 4.5. Histology and Histomorphometry

Isolated mouse tibiae or femora were soaked in 4% PFA on a shaker for 16 h, then soaked in 14% EDTA decalcification buffer (pH = 7.2) for 21 consecutive days, followed by being embedded in paraffin and then sectioned at 7 µm paraffin sections. All histology straining was performed according to the supplied protocol (Solarbio (Beijing, China), G1120, G1340, G1371, G1492). Stained tissue sections were scanned using a whole-slide imaging system (Olympus, VS200, 20× objective lens). The mice were harvested at a specified time.

### 4.6. MicroCT

Isolated mouse femora were soaked in 4% PFA on a shaker for 16 h, then soaked in PBS until analyzed by microCT (Scanco Medical AG (Brüttisellen, Switzerland), µCT45). The specific parameters referred to the previous report [38].

### 4.7. Serum CTX-I Assays

For the serum type I collagen cross-linked C-telopeptide (CTX-1) analysis, venous blood was centrifuged at 13,000× *g* in an anticoagulant serum separator gel for 15 min to collect serum, which was stored at −80 °C until use. CTX-1 analysis was accomplished with an ELISA Kit (Elabscience (Wuhan, China), E-EL-M3023).

### 4.8. Proliferation Assay

The intraperitoneal injection of EdU (APE×BIO, Houston, TX, USA), B8337) (10 μg/g body weight, dissolved in DMSO) was administered 4 h before the harvest. Subsequently, frozen sections or cells treated with 4% PFA were used to induce a click reaction using the Click-iT EdU kit (Invitrogen, C10337).

### 4.9. Cell Culture

An M2-10B4 cell line (ATCC^®^ (Manassas, VA, USA), CRL-1972™), established from BALB/c mouse bone marrow stroma, was cultured in RPMI1640 (Gibco (Waltham, MA, USA), 11875-085) with 10% FBS (ZETA life (New York, NY, USA), Z7185FBS-500) and 1% pen strep (Gibco (Waltham, MA, USA), 15070063). The cells were planted in dishes at the density of 1 × 10^4^ cells per cm^2^ 48 h prior to the experiment. To achieve the expression of Flag-tagged Gli2 proteins in the M2 cells, the full-length cDNA of mouse Gli2, including both the wild-type and S355A mutant variants, was inserted into the Fuw-tetO vector (Addgene (Cambridge, MA, USA), plasmid 20723). Subsequently, retroviruses were generated according to the lipo8000 (Beyotime (Shanghai, China), C0533) instructions. Next, M2-10B4 cells were co-infected with the FuW-rtTA virus (Addgene, plasmid 20342) and the Fuw-tetO-FlagGli2 virus. To trigger the expression of Gli2, 100 nM doxycycline (MedChemExpress (Monmouth Junction, NJ, USA), HY-N0565) was added to the culture media. shRNA targeting vectors were obtained from Tsingke. Purmorphamine (Millipore (Burlington, MA, USA), 540223), Torin1 (Tocris Biosciences (Minneapolis, MN, USA), 4247), and Cyclopamine (MedChemExpress (Monmouth Junction, NJ, USA), 4449-51-8) were purchased.

### 4.10. Western Blot Analyses

To analyze the protein abundance, RIPA (APE×BIO, K1020) was used to extract proteins from the cell samples. SDS-PAGE separation and membrane transformation of the proteins were performed by Mini-PROTEAN Tetra Cell Systems (Bio-Rad); 5% skimmed milk was used to block and dilute the antibodies, and the antibodies used for the incubation were as follows: RL2 (Invitrogen, MA1-072, 1:1000), GFAT1 (Abcam, ab125069, 1:1000), Flag (Merck, F1804, 1:1000), OGT (Abcam, ab177941, 1:1000), OGA (Abcam, ab124807, 1:1000), β-Actin (Beyotime, AA128, 1:1000), Goat anti-Mouse IgG Secondary Antibody HRP conjugated (Signaling Antibody (Beijing, China), L3032, 1:5000), and Goat anti-Rabbit IgG Secondary Antibody HRP conjugated (Signaling Antibody, L3012, 1:5000). The ultrasensitive ECL reagent kit (Beyotime, P0018M) was used to detect protein bands through chemiluminescent substrate reactions, followed by exposure in a ChemiDoc MP Imaging System (Bio-Rad (Hercules, CA, USA), 12003154) to visualize the immunoblot signals. ImageJ software, version 1.54f was employed for the quantitative densitometric analysis of the protein band intensities.

### 4.11. Co-Immunoprecipitation

Anti-Flag Immunomagnetic bead (Selleck (Houston, TX, USA), B26101) was used to precipitate out fusion-expressed proteins.

### 4.12. Quantitative PCR

TRIzol (Invitrogen, 15596018CN) was used to purify the RNA. CDNA was prepared using reverse transcription kit (Vazyme (Nanjing, China), R323-01), and the mRNA abundance was detected by real-time PCR (qPCR) using SYBR (Vazyme, Q712-02/03) with a Bio-Rad CFX96 instrument. The primer sequences refer to our previous reports [23].

### 4.13. O-GlcNAc Sites Prediction

The YinOYang website (http://www.cbs.dtu.dk/services/YinOYang/, version 1.2 (accessed on 13 March 2025)) was used to predict the O-GlcNAc sites on Gli2.

### 4.14. Statistical Analyses

For each experiment, independent biological repeats ≥3. An unpaired Student’s *t*-test or one-way ANOVA were used to calculate statistical significance. A *p*-value < 0.05 was considered to be significant.

## 5. Conclusions

The in vivo experiments demonstrated that OGT deletion affected the proliferation and osteodifferentiation of Gli1^+^ MSCs, resulting in reduced bone mass in mice. Also, a lack of OGT delayed the fracture repair in the mice. In vitro experiments revealed that Hh modulated O-GlcNAcylation through the Igf-mTORC2 signaling cascade. In addition, S355 was a key O-GlcNAc site for the Gli2 protein stabilization and osteodifferentiation of the MSCs.

## Figures and Tables

**Figure 1 ijms-26-02712-f001:**
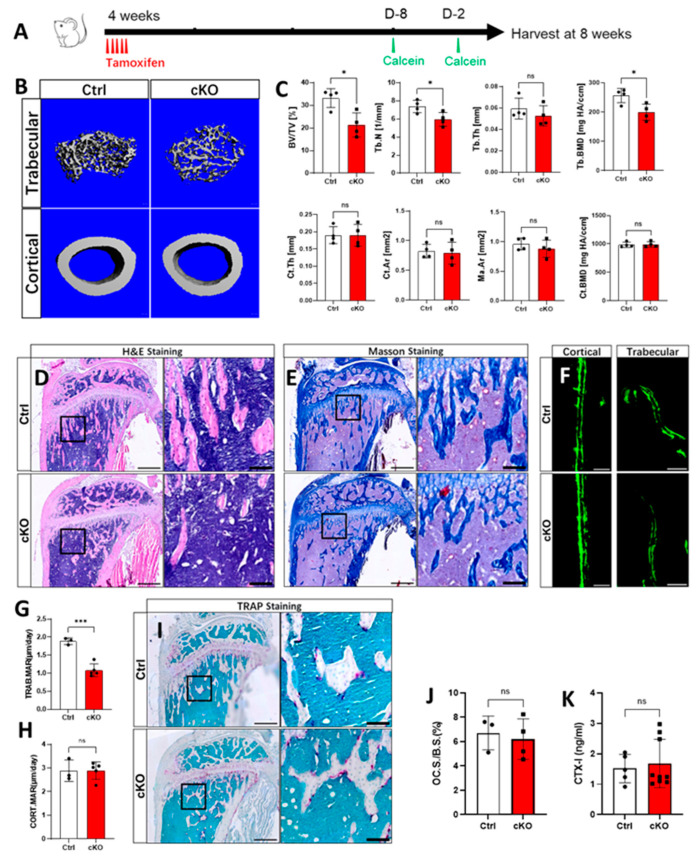
Lack of OGT in the Gli1^+^ MSCs decreased trabecular bone formation in postnatal mice. (**A**) Schematic diagram of the dosing and harvest timepoint. Two days before the sample harvest (D-2). Eight days before sample harvest (D-8). (**B**) µCT images of trabecular and cortical bone of femur. Scale bar: 100 µm. (**C**) Quantification of BV/TV, Tb.Th, Tb.N, Tb.Sp, Tb.BMD, Ct.Th, Ct.Ar, Tt.Ar, Ma.Ar, Ct, and BMD by µCT. Error bar: mean ± SD. ns, not significant, * *p* < 0.05, *n* = 4, Student’s *t*-test. (**D**,**E**,**I**) The H&E, Masson, and TRAP stainings of longitudinal sections through the proximal tibia. Scale bar: 500 µm. Boxed areas in (**D**,**E**,**I**) are shown at a high magnification to the right. Scale bar: 100 µm. (**F**) Representative images of double labeling for bone formation on the endosteal surface of the tibia. Green: calcein. Scale bar: 10 µm. (**G**,**H**) Quantification of the mineral apposition rate (MAR) at the endosteal bone surface from double-labeling experiments. Error bars: mean ± SD. *** *p* < 0.001, ns, not significant, *n* ≥ 3, Student’s *t*-test. (**J**) Osteoclast (TRAP-positive multinucleated cells) surface normalized to bone surface (OC. S./B. S.) is shown. Error bars: mean ± SD. ns, not significant. *n* ≥ 3, Student’s *t*-test. (**K**) Serum level of P1NP at the time of harvest. Error bars: mean ± SD. ns, not significant. *n* ≥ 5, Student’s *t*-test.

**Figure 2 ijms-26-02712-f002:**
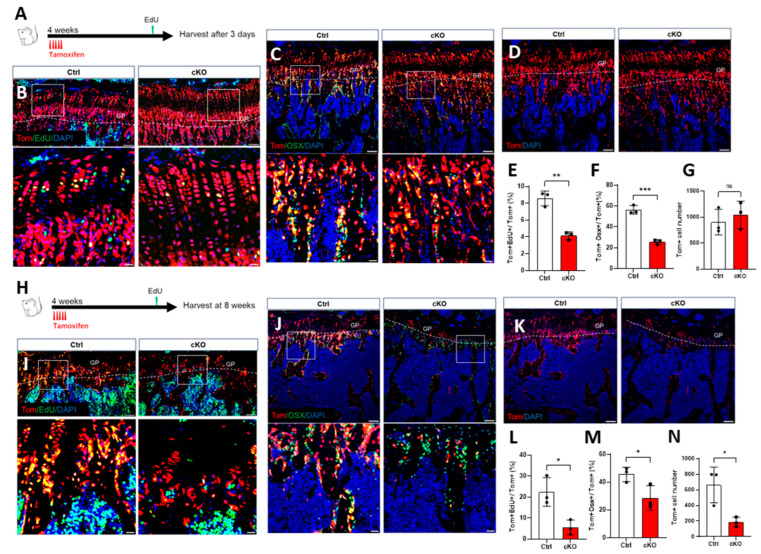
OGT depletion led to the depletion of the Gli1^+^ MSCs pool. (**A**,**H**) Schematic diagram of the dosing and harvest time point for the chasing experiment. (**B**–**G**) Mice with the genotypes Gli1-CreERT2; OGT^+/Y^; Ai9 (Ctrl) and Gli1-CreERT2; OGTc/Y; Ai9 (cKO) were treated with TAM at 4 weeks of age for five consecutive days and harvested after 3 days. (**B**,**C**) Representative confocal images of the primary spongiosa below the growth plate in the proximal tibia showing colocalization of tdTomato with EdU (**B**) or Osx (**C**). GP: growth plate. Scale bar: 100 µm. Boxed areas are shown below at a higher magnification. Scale bar: 20 µm. The dotted line indicates the boundary between the growth plate and the primary spongiosa. Same below. (**D**) Representative confocal images showing the direct fluorescence of tdTomato-labeled Gli+ cells. (**E**–**G**) Quantification of the ratio of Tom+EdU+ over Tom+, the ratio of Tom+Osx+ over Tom+, and the number of Tom+ cells in the primary spongiosa the area within 300 µm and spanning the width of the bone flanked by the periosteum under the growth plate. (**I**–**N**) Ctrl and cKO mice were treated with TAM at 4 weeks of age for five consecutive days and chased for one additional month. (**I**,**J**) Representative confocal images of primary spongiosa below the growth plate in the proximal tibia showing colocalization of tdTomato with EdU (**I**) or Osx (**J**). (**K**) Representative confocal images showing direct fluorescence of tdTomato-labeled Gli+ cells. (**L**–**N**) Quantification of the ratio of EdU+OSX+ over Tom+, the ratio of Tom+OSX+ over Tom+, and the number of Tom+ cells. *n* = 3. Error bars: mean ± SD. ns, not significant, * *p* < 0.05, ** *p* < 0.01, *** *p* < 0.001. *n* = 3, Student’s *t*-test.

**Figure 3 ijms-26-02712-f003:**
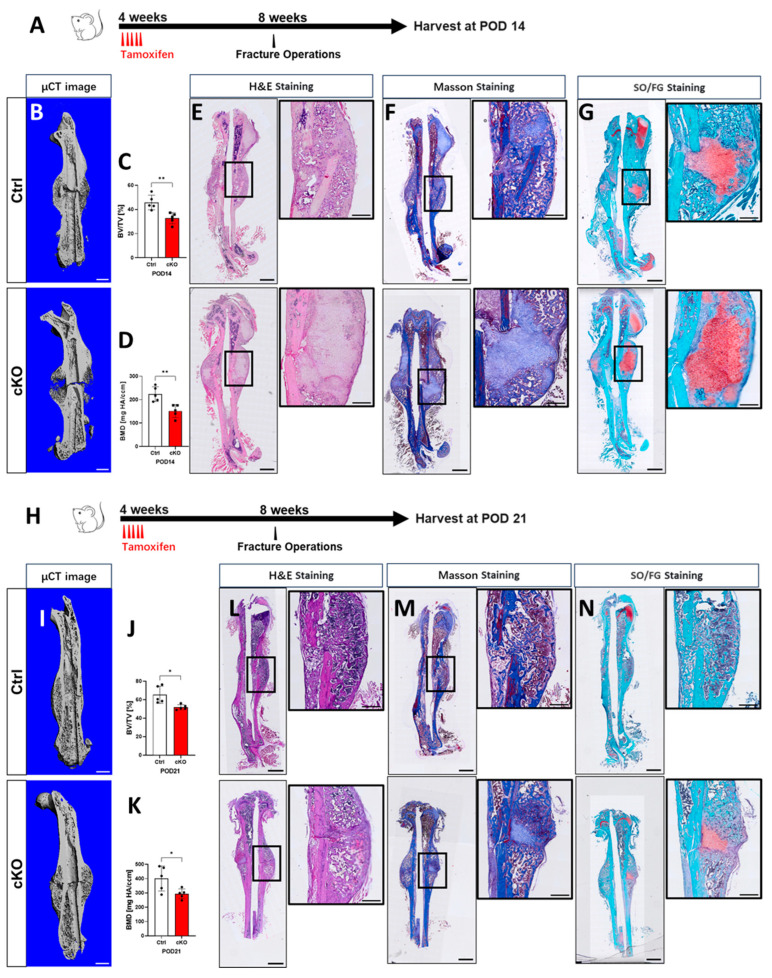
OGT deletion in Gli1^+^ MSCs delayed the fracture repair. (**A**,**H**) Schematic diagram of the dosing, operation, and harvest time point for fracture repair. (**B**–**G**) POD 14 and (**I**–**N**) POD 21. *n* = 5. (**B**,**I**) Representative micro-CT 3D reconstruction images. Scale bar: 1 mm. (**C**,**D**,**J**,**K**) Micro-CT analysis of the femoral callus at POD 14 (**C**,**D**) and POD 21 (**J**,**K**). Data are shown as the mean ± SD. Error bars: mean ± SD. * *p* < 0.05, ** *p* < 0.01. Student’s *t*-test. BMD: bone mineral density; BV/TV: bone volume over tissue volume. Data were acquired from the slices of the whole femoral callus. (**E**–**G**,**L**–**N**) Representative images of the (**E**,**L**) H&E-, (**F**,**M**) Masson-, and (**G**,**N**) SO/FG-stained sections of the medial sections through the fractured femur at POD 14 and POD 21. Scale bar: 1 mm. The boxed areas in (**E**–**G**,**L**–**N**) are shown at high magnification to the right. Scale bar: 300 µm.

**Figure 4 ijms-26-02712-f004:**
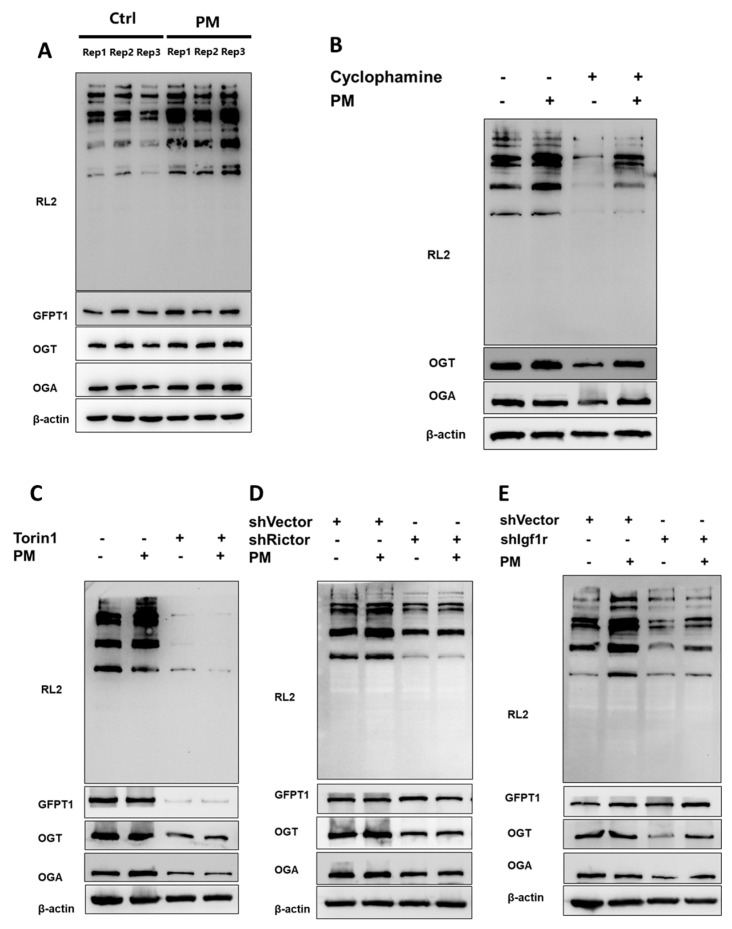
Hh modulated O-GlcNAcylation through the Igf-mTORC2 signaling cascade. (**A**) M2 cells were treated with 1 µM PM for 7 days, and the O-GlcNAc levels were detected by Western blotting. Representative Western blots of RL2, GFPT1, OGT, and OGA in the M2 cells are shown, *n* = 3, same below. (**B**) Western blot after 1 µM Cyclophamine or PM treatment for 7 days. (**C**) Western blot after 250 nM Torin pre-treatment for 24 h and PM treatment for 7 additional days. (**D**) Western blot after shRNA knockdown Rictor and PM treatment for 7 days. (**E**) Western blot after shRNA knockdown Igf1r and PM treatment for 7 days.

**Figure 5 ijms-26-02712-f005:**
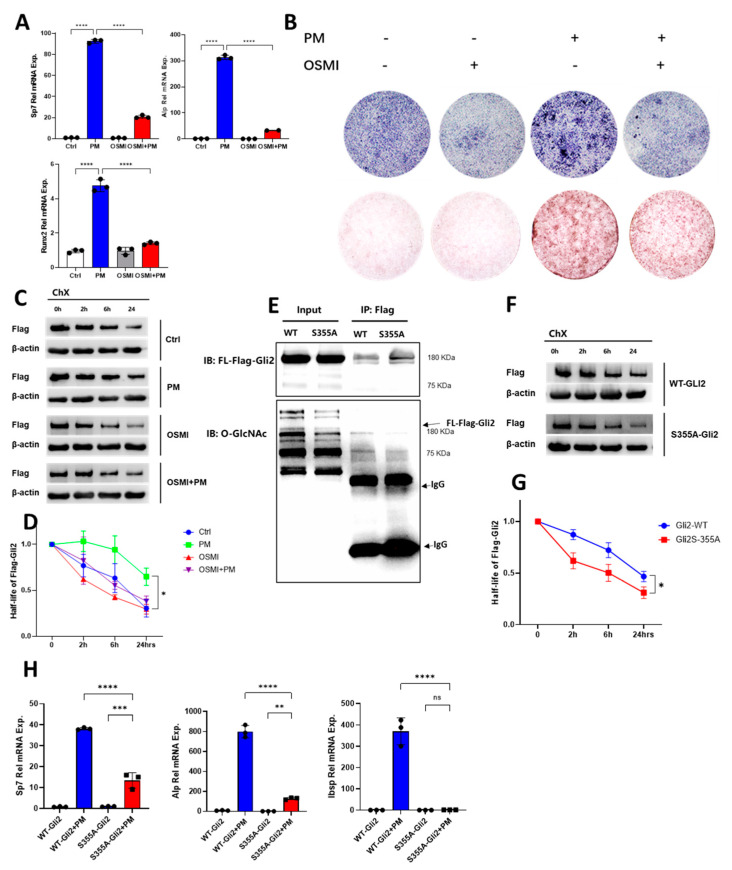
O-GlcNAcylation of Gli2 was necessary for Hh-induced osteogenesis. (**A**) qPCR analyses in M2 cells pre-treated with 20 µM OSMI before PM treatment for 7 days. (**B**) ALP staining and Alizarin Red S staining were performed after PM administration for 3 and 21 days. (**C**,**D**) Effects of PM and OSMI on the Flag-PDK1 turnover, as measured by Western blotting (**C**). The quantifications are shown (**D**). * *p* < 0.05, two-way ANOVA followed by Tukey’s multiple comparisons test, *n* = 3. (**E**) IP of Flag-Gli2-S355A and detection with O-GlcNAc immunoblotting. (**F**,**G**) Effects of S355A on the half-life of Flag-Gli2, as measured by Western blotting (**F**) and presented with quantification (**G**). * *p* < 0.05, two-way ANOVA followed by Tukey’s multiple comparisons test, *n* = 3. (**H**) qPCR analyses in S355A and WT cells in response to PM for 3 days. *n* = 3. Error bars: mean ± SD. ns, not significant, ** *p* < 0.01, *** *p* < 0.001, **** *p* < 0.0001, * *p* < 0.05; one-way ANOVA followed by Tukey’s multiple comparisons test.

**Figure 6 ijms-26-02712-f006:**
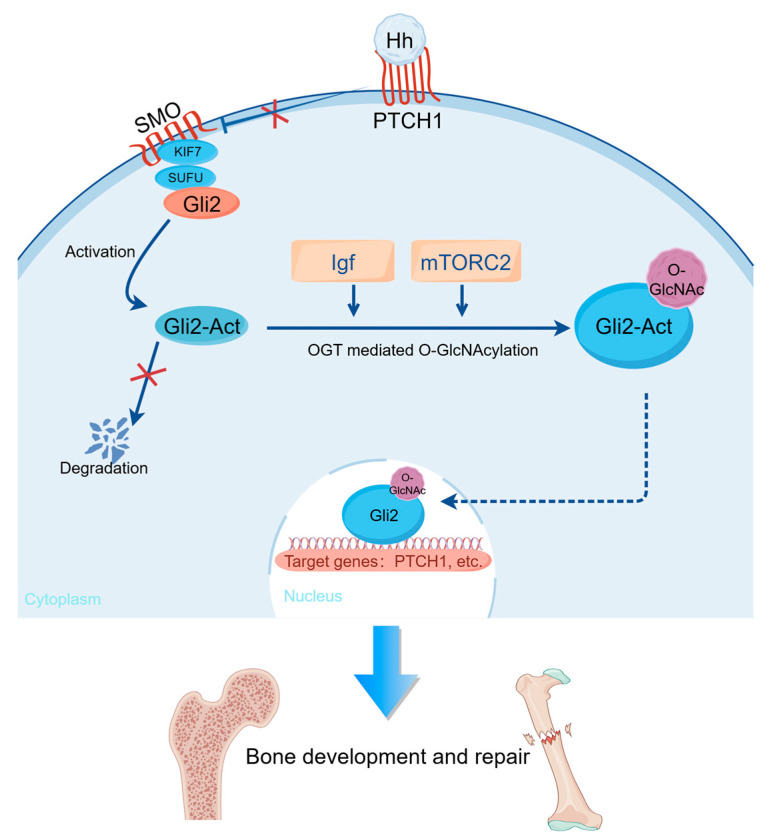
Schematic of how Hh modulates O-GlcNAcylation’s promotion of bone formation and repair through the Igf-mTORC2 signaling cascade. Schematic was created with FigDraw.

## Data Availability

All the raw data supporting the conclusions of this article will be made available by the authors upon request.

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
