# Peer review of "O-GlcNAcylation in Gli1+ Mesenchymal Stem Cells Is Indispensable for Bone Formation and Fracture Healing"

_ijms, 2025, doi:10.3390/ijms26062712_

Round 1
Reviewer 1 Report
Comments and Suggestions for Authors
A well-structured study design and discussion aimed at elucidating the hedgehog signaling pathways, including Ptch receptors, Hh ligand, Smoothened, Gli2, Gli1, MSC cells, O-GlcNAcylation, and the role of O-GlcNAc glycosyltransferase (OGT) in bone health and fracture repair. Please see the comments below for further consideration.
- Recommend including a conclusion section.
- Referencing error found on line 107.

Reviewer 2 Report
Comments and Suggestions for Authors
This manuscript sheds light on the important role of O-GlcNAcylation in Gli1+ mesenchymal stem cells (MSCs) and how it influences bone formation and fracture healing. The study is well-designed, combining genetic models, in vivo imaging, and molecular analyses to explore the link between Hedgehog (Hh) signaling and O-GlcNAcylation in osteogenesis. The findings add valuable knowledge to the field of bone biology and could have exciting implications for stem cell-based regenerative medicine. However, several questions need further clarification:
- In Figure 1, the sample size varies between tests (some have 5 samples, while others have 3). Is there a specific reason for this discrepancy?
- Were all samples assessed through multiple measurements? If so, what accounts for the differences in sample numbers across different tests?
- The manuscript states the use of both female and male, did you see any differences between them? Were statistical comparisons made between male and female groups to confirm this? If sex differences were not analyzed separately, would pooling data from both sexes introduce variability in the results?
- In line 283, the manuscript states that phosphorylation and O-GlcNAcylation compete for the same serine or threonine residues. Given this, do Gli2 phosphorylation patterns change upon OGT depletion?
- The study suggests that OGT deletion delays fracture healing. Would it be possible to test whether OGT overexpression in Gli1+ MSCs enhances fracture healing?
The quality of English in the manuscript is generally good, with clear scientific communication and well-structured arguments.
Reviewer 3 Report
Comments and Suggestions for Authors
- Abstract: What do Igf & OGT stand for?
- Figure 1A: The figure is too simple and lacks of explanation of the key points and authors are suggested to add more details about imaging, MSC isolation.
- Page 3 lines 107&108: (Figure 2Error! Reference source not found.A)Er- 107 error! Reference source not found..what does it mean?
- So may abbreviations in the figure legends need to be explained.
- - In western blot data: The authors are recommended to provide arrows to the specific bands.
- What is the role of M2-10B4 cells in this study?
- The authors mentioned MSCs....what type of MSCs, Isolation, and characterization information are lacking?
- The authors mentioned the link to Wnt signaling: Any supporting data?
- The authors are recommended to provide conclusion schematic diagram that summarizes their findings.
- Overall, the topic is interesting, however, the data quality, methods, and conclusion is too poor and need substantial revision.
Need to improve the present grammatical errors and typos.
